# The impact of the COVID-19 pandemic on Italian population-based cancer screening activities and test coverage: Results from national cross-sectional repeated surveys in 2020

Paolo Giorgi Rossi[1], Giuliano Carrozzi[2], Patrizia Falini[3], Letizia Sampaolo[2], Giuseppe Gorini[3], Manuel Zorzi[4], Paola Armaroli[5], Carlo Senore[5], Priscilla Sassoli de Bianchi[6], Maria Masocco[7], Marco Zappa[8], Francesca Battisti[3], Paola Mantellini[3,8]*

[1]Azienda Unità Sanitaria Locale - IRCCS di Reggio Emilia, Reggio Emilia, Italy; [2]Azienda Unità Sanitaria Locale Di Modena, Modena, Italy; [3]Istituto per lo Studio, la Prevenzione e la Rete Oncologica (ISPRO), Florence, Italy; [4]Registro Tumori del Veneto, Azienda Zero, Padua, Italy; [5]Centro di Prevenzione Oncologica, Azienda Ospedaliero-Universitaria Città della Salute e della Scienza di Torino, Turin, Italy; [6]Servizio Prevenzione Collettiva e Sanità Pubblica, Direzione Generale Cura della Persona, Salute e Welfare, Regione Emilia-Romagna, Bologna, Italy; [7]Istituto Superiore di Sanita, Rome, Italy; [8]Osservatorio Nazionale Screening, Florence, Italy

*For correspondence:
p.mantellini@ispro.toscana.it

Competing interest: The authors declare that no competing interests exist.

## Abstract

**Background:** In Italy, regions have the mandate to implement population-based screening programs for breast, cervical, and colorectal cancer. From March to May 2020, a severe lockdown was imposed due to the COVID-19 pandemic by the Italian Ministry of Health, with the suspension of screening programs. This paper describes the impact of the pandemic on Italian screening activities and test coverage in 2020 overall and by socioeconomic characteristics.

**Methods:** The regional number of subjects invited and of screening tests performed in 2020 were compared with those in 2019. Invitation and examination coverage were also calculated. PASSI surveillance system, through telephone interviews, collects information about screening test uptake by test provider (public screening and private opportunistic). Test coverage and test uptake in the last year were computed by educational attainment, perceived economic difficulties, and citizenship.

**Results:** A reduction of subjects invited and tests performed, with differences between periods and geographical macro areas, was observed in 2020 vs. 2019. The reduction in examination coverage was larger than that in invitation coverage for all screening programs. From the second half of 2020, the trend for test coverage showed a decrease in all the macro areas for all the screening programs. Compared with the pre-pandemic period, there was a greater difference according to the level of education in the odds of having had a test last year vs. never having been screened or not being up to date with screening tests.

**Conclusions:** The lockdown and the ongoing COVID-19 emergency caused an important delay in screening activities. This increased the preexisting individual and geographical inequalities in access. The opportunistic screening did not mitigate the impact of the pandemic.

**Funding:** This study was partially supported by Italian Ministry of Health – Ricerca Corrente Annual Program 2023 and by the Emilian Region DGR 839/22.

## Editor's evaluation

The authors provide important evidence for the impact of the COVID-19 pandemic on screening for breast, cervix, and colorectal cancer in Italy. They compared Invitation and examination coverage, conducted telephone interviews, and investigated screening test coverage, before and during the pandemic, according to educational attainment, perceived economic difficulties and citizenship. Their findings convincingly show that the lockdown and pandemic restrictions caused delays in screening and increased the pre-existing individual and geographical inequalities in access.

## Introduction

Since early 2020, the COVID-19 pandemic and the measures taken by most governments to control the spread of the virus had an impact on all health services, but also on people's behaviors and attitudes toward prevention (*Kumari et al., 2021*; *Moynihan et al., 2021*). The combination of reduced health service delivery for non-COVID-19 activities and a lower propensity to access health services by the population caused appreciable delays in cancer diagnosis in most countries where the phenomenon has been studied (*Ferrara et al., 2021*; *Dinmohamed et al., 2020*; *Nyante et al., 2021*).

Cancer screening tests are nonurgent services, and thus they were among the first suspended activities during the first pandemic wave in most European countries (*Figueroa et al., 2021*). On the other hand, organized screening programs actively invite the target population. The active invitation allows to accurately plan the workload, which represented an opportunity for organized screening programs to resume post-lockdown activities in a rational way according to accurate prioritization, aiming to minimize the impact of the pandemic on cancer diagnosis delays (*Castanon et al., 2021*, *Campbell et al., 2021*). Thus, the presence of a structured and well-organized program has been recognized as a possible element favoring the resilience of health services to the pandemic disruption (*Mangone et al., 2022*).

In Italy, a national law included organized screening programs for breast, cervical, and colorectal cancers among the public health interventions that all the regions must carry out (*Decreto del Presidente del Consiglio dei Ministri, 12 gennaio, 2017*). The target population, the test, and the intervals used are reported in *Table 1*. Before the COVID-19 pandemic, the invitation coverage was almost complete for all screening programs in Central Italy, and for breast cancer in Northern Italy, while for colorectal cancer screening, there were still areas, especially in Southern Italy, where large parts of the target population were not actively invited. There are large differences in participation in all three screening programs among regions, with the northern regions achieving higher participation rates than the southern ones. Routine statistics on activity and performance indicators are produced by the National Screening Monitoring Center (ONS), which is a technical network appointed by the Italian Ministry of Health to monitor regional screening campaigns, and they are available at https://www.osservatorionazionalescreening.it/.

Across the country, opportunistic screening – offered by both private and public providers – is common and does not have a specific informative flow for reporting and monitoring. In 2010–13

**Table 1.** Italian Ministry of Health recommendations for cancer screening programs.

|  | Target population | Test | Interval |
| --- | --- | --- | --- |
| Cervical cancer | Women 25–64 years | Pap test (25 to 29/34 years) HPV test (30/35–64 years) | 3 years after negative Pap 5 years after negative HPV |
| Breast cancer | To be implemented: Women 50–69 years Suggested: Women 45–49 years Women 70–74 years | Mammography | 2 years for women 50–74 1 year for women 45–49 |
| Colorectal cancer | To be implemented: Women and men 50–69 years Suggested: Women and men 70–74 years | Fecal immunochemical test (FIT) or Sigmoidoscopy at 58/60 years | 2 years after negative FIT Once in a life sigmoidoscopy |

**Figure 1.** Cumulative incidence (left) and mortality (right) rates in the first (upper panel, March–June 2020) and second COVID-19 wave (lower panel, July–December 2020) per 100,000 inhabitants. Mortality is referred to the date of incidence. Rates are computed by province, bold lines define the macro areas, North, Center, and South, and Islands. Data from the National Institute of Health, Italy, 2020.

in Italy, test coverage in the target population that reaches 75, 80, and 48% for breast, cervical, and colorectal cancers, respectively. The contribution of opportunistic screening to coverage differed across the three screening programs with about one fourth, one third, and one sixth of the coverage attributable to opportunistic testing for breast, cervical, and colorectal cancer screening test coverage, respectively (*Carrozzi et al., 2015*).

In Italy, the first diagnosis of COVID-19 was made on February 20, 2020, and a strict lockdown started on March 8 (*Marziano et al., 2021*). The impact of this first wave in terms of deaths was very strong and concentrated in Northern Italy. A second wave started in October and lasted until the end of the year, involving all the Italian regions (*Figure 1*). Control measures differed in the three periods: from March to May, the lockdown stopped all nonessential activities; during the summer, almost all

restrictions were removed; while during the October to December restrictions, school closures, limits to movement and recommendations to work from home were applied on a regional or even provincial basis according to incidence (*Manica et al., 2021*).

The aim of this paper is to describe the impact of the pandemic and infection control measures on the activities of Italian screening programs in terms of invitations and screening tests performed during the first year of the pandemic and to investigate how this affected the population screening test coverage overall and by socioeconomic characteristic of the target population.

## Methods

### Setting and description of the infection control measures

In Italy, breast, cervical, and colorectal cancer screenings are recommended, and regional health systems are in charge of implementing them according to the recommendations of the European Commission and of the Italian Ministry of Health. The target ages, intervals, and test modalities recommended in Italy are reported in *Table 1*; *Ministero della Salute e Osservatorio Nazionale Screening, 2006*, *Ronco et al., 2012*.

After the first COVID-19 case was diagnosed on February 20, apparently, small clusters were identified and restrictions on movements in small areas in Northern Italy were set. On March 9, the first lockdown measures were put in place for the whole country, causing the suspension of screening first-level activities. On the contrary, national directives recommended maintaining diagnostic assessment in those who tested positive and assuring all oncological follow-up (*Decreto del Presidente del Consiglio dei Ministri del 9 marzo, 2020*, *Ministero della Salut, 2020*).

The strict lockdown, that is, the 'stay at home' period in which only essential activities were allowed, ended at the beginning of May 2020, but the restrictions were gradually removed until the beginning of June 2020, when only physical distancing and wearing face masks remained mandatory (*Marziano et al., 2021*). During the summer, COVID-19 incidence remained relatively low throughout the country, but in October it increased rapidly and new restrictions were introduced (*Manica et al., 2021*). Regions or provinces were classified as white, yellow, orange, and red according to a set of indicators measuring the quality of data reporting, the testing capacity, the incidence trend (the Rt), the adequacy of contact tracing, and the pressure on the health system (*Riccardo et al., 2022*). Each color code corresponded to a set of mandatory restrictions that the regional government should implement and eventually integrate with local measures. Among these measures, none was directed to reduce nonurgent health services and, in several regions, cancer screening had been included among the services, which had to be maintained. Nevertheless, in many areas, the pressure on hospitals became so strong that it became necessary to reduce nonurgent activities in order to redirect health professionals to COVID-19-related activities. Furthermore, in orange and red zones there were restrictions on moving from one municipality to another (even if these did not apply for medical checks/reasons) and restrictions on public transport. *Figure 1* summarizes the COVID-19 incidence and mortality in Italy by geographical area during the first and the second waves in 2020.

### Study design

This study presents the results of two national surveys. The first collected the screening activities, in terms of invitations and first-level tests performed, of the public, organized screening programs during 2020 and the first five months of 2021, compared with the same activities in performed 2019. The second survey is the PASSI's survey (one of the two Italian National Health Interviews), which collects information on screening uptake by the target population, both in organized screening and in opportunistic screening.

From the first survey, we can assess how much the screening activities were slowed down by the pandemic and the magnitude of the backlog and consequent delay in screening the target population.

From the second survey, we measure the impact of the pandemic on test coverage in the target population, and the proportion of the target population who had a test in the last year. From this source, we can distinguish the tests performed in public programs and in private opportunistic screening, and we can also measure the coverage by socioeconomic characteristics of the target population.

## Data sources

The National Screening Monitoring Center (ONS) monitors regional screening performances and trends, and a summary report is regularly published (https://www.osservatorionazionalescreening.it/content/rapporto-ons-2020). In October 2020, the ONS promoted an additional survey to monitor the impact of the pandemic on screening programs (*Mantellini et al., 2020*; *Battisti et al., 2022*).

An ad hoc qualitative and quantitative questionnaire was sent by the ONS to all regional cancer screening coordinators. The qualitative part included the description of the changes in screening activities adopted during the pandemic period, including the suspended activities (i.e., invitations, spontaneous access, second level tests, and assessments) and chances in invitation pace. The quantitative part collected, for the three screening programs, the absolute number of subjects invited and the absolute number of screening tests performed for the periods of January–May 2020, June–September 2020, October–December 2020, and January–May 2021 compared to those of the same periods over 2019.

Data were referred to the core target population, that is, the age group that all regions must implement (see *Table 1*).

PASSI's survey is one of the two National Health Interviews (NHIS) active in Italy (*Carrozzi et al., 2015*; *Petrelli et al., 2018*). Through a continuous sampling of the population aged 18–69 residing in Italy, it conducts telephone interviews collecting information about health behaviors, health conditions, socioeconomic conditions, use of health services, and participation in preventive interventions (*Baldissera et al., 2011*). Sampling methods are described elsewhere; briefly a nationally representative sample stratified by age, sex, and local health authority is drawn. Participation in the survey is free and voluntary, individuals can refuse to be interviewed; the average response rate was 80% in the period 2017–2020. Nonparticipants are substituted from a list of subjects of the same stratum (*Baldissera et al., 2014*). The interviewers are specifically trained to process personal data safely and correctly. Individuals selected for the interview are informed by letter about the objectives of the investigation, its methods, and the arrangements taken to ensure the confidentiality of the collected information. After receiving the letter, they are contacted by phone; during the phone interview, the interviewer presents the information again and asks for the interviewee's consent to conduct the interview.

In this study, the analyzed data were collected by PASSI between 2017 and 2020, including 44,874 (of which 6736 conducted in 2020) interviews of women aged 25–64 years informing on cervical cancer screening, 23,276 (of which 3501 conducted in 2020) interviews of women aged 50–69 years informing on breast cancer screening, and 40,826 (of which 6233 conducted in 2020) interviews of women and men aged 50–69 years informing on colorectal cancer screening. Lombardy region suspended the surveillance in 2016. For colorectal cancer screening, data from the Piedmont region are excluded from analyses using tests performed in the last year as the outcome because organized screening programs offer a flexosigmodoscopy once in life as the primary test.

PASSI provides information on test coverage in the target population, including both the share of tests performed within the organized screening programs and those performed outside (spontaneous screening). PASSI provides data on the differences in the execution of screening tests also with respect to socio-demographic characteristics. The exact number of interviews included in the analyses for each question is reported in *Supplementary file 7*.

## Outcomes definition

Based on the ONS survey, we report the number of invitations sent during the investigation period and the number of screening examinations performed in the study period. Invitation (percentage of citizens who were sent an invitation to a screening during the analyzed period compared to the population to be invited in the period in order to reach all the target population in the screening interval, excluding undelivered invitations and noneligible subjects) and examination (percentage of citizens who performed the test compared to the population to be tested in the period in order to reach all the target population in the screening interval, excluding those with specific exclusion criteria) coverage relatively to 2017–2019 is also reported.

We also computed the 'standard months' of delay, that is, the number of months that would be required to catch up with the cumulated backlog if the program screened women at the same pace, as it did over the pre-COVID era. This parameter is obtained by multiplying the reduction in the number

of tests performed during the study period compared to the same period in 2019 (% reduction) by the duration (number of months) of the study period.

Based on the date of the last test before the PASSI interview and the reported provider of the last test (free or paid out of pocket, proxy of organized and spontaneous screening, respectively), we computed the test coverage for each screening program: for breast cancer, we considered as being eligible the female population aged 50–69 years and those who reported having had a mammogram in the last 2 years as up to date with screening; for cervical cancer, we considered as being eligible the female population aged 25–64 years and those having had a Pap test in the last 3 years or an HPV-DNA test in the last 5 years as up to date with screening; for colorectal cancer, we considered as being eligible males and females aged 50–69 years and those reporting a fecal occult blood test (FOBT) in the last 2 years or a colonoscopy or sigmoidoscopy in the last 5 years as up to date with screening.

We also only considered the tests performed in the last year as an outcome for each screening test.

## Statistical analysis

For the ONS surveys, only descriptive analyses are presented.

In PASSI, each Local Health Authority extracts a proportionate stratified sampling for the sex and age categories (18–34, 35–49, and 50–69 years) of the resident population. Therefore, data analysis at a national and macro-area level requires the application of appropriate weights accounting for age and geographic stratification to be representative of the whole population.

Trends of coverage are computed for each quarter of the study period, including interviews from January 2008 up to December 2020 for cervical and breast cancer and from January 2010 to December 2020 for colorectal cancer screening because the relevant items in the questionnaire were changed in 2010.

Using the tests performed in the last year as a dependent variable, we present Poisson regression models reporting the odds of having had a test in the last year vs. the odds of not having the test in the last year. Prevalence rate ratios with the relative 95% CI for age, gender, educational attainment (four categories: elementary school; middle school; high school; higher education), nationality (two categories: Italians or foreign nationals from high-income countries; foreign nationals from middle- or low-income countries - according to the World Bank classification [UNDP, 2007]) and economic difficulties (three categories: many economic difficulties; some economic difficulties; no economic difficulties) are obtained. Models are performed on interviews conducted in 2020 and for those conducted in the 2017–2019 period. No formal tests of hypothesis have been performed and no predefined significance threshold has been fixed in this study, 95% CI boundaries should be interpreted as continuous variables.

The statistical package Stata 16 software (StataCorp LP) was used to analyze the data.

## Results

### Impact on screening programs

In total, 21 regions out of 21 participated in the survey. In one region, Calabria, only data from three out of five provinces were available; the data from Basilicata refer to the whole period of the study, thus it is excluded from sub-period analyses; the colorectal cancer screening data from Umbria refer to the 50–74-year-old target population rather than 50–69.

With the first lockdown measures on March 9, 2020, all screening first-level activities should be suspended maintaining diagnostic assessment in those who tested already positive. Qualitative data from the survey show that, regardless of national directives, the suspension was heterogeneous. It was almost complete in most Northern and Central regions where screening invitations and test delivery were immediately suspended; in Lazio, the suspension was established late; while in other regions, according to the screening organization, test delivery was maintained for colorectal (Puglia, Umbria) and cervical (Valle D'Aosta) cancer campaigns. Assessment of people who had previously had a positive screening test was never stopped in any program. Most screening programs started again in May/June, but rules to reduce the risk of infection required avoiding crowding in waiting rooms and physical distancing in the clinics, thus the number of exams per hour was reduced by 30–50% in all programs. These restrictions lasted for the entire study period. Furthermore, many programs reported a reduction in the pace of invitations during the second wave of the pandemic in the autumn of 2020.



**Figure 2.** Invitation and examination coverage for cervical, breast, and colorectal cancer screening in Italy, by year and geographical macro area. The invitation coverage (right panel) is computed as the number of invitations sent during the year divided by the expected target population to be invited in 1 year. Test coverage (right panel) is computed as the number of tests performed during the year divided by the expected target population in that year. For breast and colorectal cancer, the target population is expected to be invited in 2 years, for cervical cancer the target population is expected to be invited in 3 years if the last test was a Pap test and every 5 years if the last test was an HPV test.

According to the quantitative survey, in 2020, the screening invitations decreased, for cervical, breast, and colorectal cancer screening in Northern and Southern Italy, compared with those of the 2017–2019 period. It is worth noting that Central Italy registered the best performances: cervical cancer screening programs were indeed able to maintain invitation coverage close to 100% and breast and colorectal cancer screening resulted in just below 90% (*Figure 2*).

The reduction in invitations was large and consistent in all macro areas and all screening programs for the first (January to May 2020) and second (June to September 2020) periods. In the third one (October to December 2020), differences emerged: in Central Italy, programs tried to catch up with the backlog of invitations, while in Northern Italy the programs mostly continued with the pre-pandemic pace. In Southern Italy, the reduction in activity remained up to the first quarter of 2021, except for colorectal cancer screening (*Figure 3*).

**Figure 3.** Changes in the number of invitations sent (left panel) and screening tests (right panel) performed by screening programs in 2020–2021 compared to the same months in 2019, by period and geographic macro area. Data from ONS survey.

Compared to 2017–2019, in 2020 the reduction in examination coverage was larger than the reduction in invitation coverage for all screenings and in all macro areas (*Figure 1*). In Central and Northern Italy, it was particularly strong in the first period and then decreased gradually (*Figure 3*), reaching pre-pandemic levels for breast and colorectal cancer screening in the first quarter of 2021, but not for cervix cancer screening in Northern Italy. In Southern Italy, the reduction in tests performed lasted until the end of 2020 and it is still strong for cervical and breast cancer screening in the first quarter of 2021 (*Figure 3*).

The delay accumulated until May 2021 in screening the target population differs by macro area, and it is larger for Southern Italy and smaller for Central Italy for the three programs. Even though the efforts in restarting invitations were dissimilar, the difference in delay between breast and cervical cancer was only 1.2 months. Ranges between regions within macro areas are important. In fact, in Northern and Central Italy one or more regions cumulated a negligible delay of fewer than 45 days, while some regions cumulated about 1 year of delay in all programs (*Table 2*).

## Impact on overall screening test coverage

The trend for test coverage as reported by PASSI showed a clear decrease in all the macro areas for the mammographic and colorectal screenings starting from the second half of 2020 (*Figure 4*). Also,

**Table 2.** Cumulative reduction of tests performed in Italian screening programs and average cumulated delay in testing, with ranges between regions, by geographical macro area.

January 2020 to May 2021. Data from ONS survey.

| Macro area | Cervix | | | | Breast | | | | Colorectal | | | |
|---|---|---|---|---|---|---|---|---|---|---|---|---|
| | Test cumulative reduction Jan 2020–May 2021 | Average delay in months | Range between regions | | Test cumulative reduction Jan 2020–May 2021 | Average delay in months | Range between regions | | Test cumulative reduction Jan 2020–May 2021 | Average delay in months | Range between regions | |
| | | | Minimum | Maximum | | | Minimum | Maximum | | | Minimum | Maximum |
| North | –409,092 | –6.4 | –12.1 | +7.5 | –438,744 | –4.5 | –10.1 | –0.9 | –800,101 | –5.9 | –14 | +2.7 |
| Center | –136,393 | –4.2 | –6.6 | –0.5 | –154,783 | –4.0 | –6.0 | –1.4 | –213,418 | –4.4 | –6.3 | –0.8 |
| South and Islands | –239,275 | –7.2 | –12.7 | –5.6 | –223,439 | –6.9 | –11.2 | –5.8 | –182,468 | –8.4 | –13.4 | –2 |
| Italy | –784,760 | –6.0 | | | –816,966 | –4.8 | | | –1,195,987 | –5.8 | | |

ONS: National Screening Monitoring Centre.



**Figure 4.** Trends of the proportion of the screening target population who declared to have had a test in due time, overall, and by the setting of the last test. Data from the PASSI interviews. For breast cancer, we considered as being eligible the female population aged 50–69 years and those who reported as having had a mammogram in the last 2 years as up to date with screening; for cervical cancer, we considered as being eligible the female

*Figure 4 continued on next page*

*Figure 4 continued*

population aged 25–64 years and those having had a Pap test in the last 3 years or an HPV-DNA test in the last 5 years as up to date with screening; for colorectal cancer, we considered as being eligible males and females aged 50–69 years and those who reported as having had a fecal occult blood test (FOBT) in the last 2 years or a colonoscopy or sigmoidoscopy in the last 5 years as up to date with screening.

for coverage with Pap tests or HPV tests, the decrease is appreciable, but the magnitude is smaller. It is also appreciable that in 2020 we had an inversion in a long-term trend, with a decrease in opportunistic screening in favor of organized screening for cervical cancer (*Figure 5*).

The decrease in test coverage is steeper in people with a lower level of educational level or with many perceived economic difficulties (*Figures 6 and 7*). For cervical cancer, the proportion of women aged 25–64 years that declared to have a test in the last year decreased dramatically for the screening program and to a lesser extent for opportunistic tests. For breast and colorectal cancer, the reduction was smaller and all attributable to organized screening (*Figure 8*).

**Table 3.** Multivariable Poisson regression models comparing the prevalence of having had a test in the last year by age, sex, familial status, socioeconomic characteristics, citizenship, and pre-pandemic and pandemic period for cervical, breast, and colorectal cancer screening in Italy. PRR: Prevalence Rate Ratio.

| | Cervix | | | Breast | | | Colorectal | | |
|---|---|---|---|---|---|---|---|---|---|
| | PRR | 95% CI | | PRR | 95% CI | | PRR | 95% CI | |
| **Age (years)** | | | | | | | | | |
| 25–34 | 1.06 | 1.01 | 1.11 | | | | | | |
| 35–49 | 1.10 | 1.06 | 1.14 | | | | | | |
| 50–64 | ref. | | | | | | | | |
| 50–59 | | | | 1.13 | 1.08 | 1.18 | ref. | | |
| 60–69 | | | | ref. | | | 1.15 | 1.10 | 1.20 |
| **Sex** | | | | | | | | | |
| Male | | | | | | | ref. | | |
| Female | | | | | | | 0.99 | 0.95 | 1.03 |
| **Familial status** | | | | | | | | | |
| Married or with partner | 1.09 | 1.05 | 1.13 | 1.03 | 0.99 | 1.08 | | | |
| Alone | ref. | | | ref. | | | | | |
| **Educational level** | | | | | | | | | |
| No title/elementary | ref. | | | ref. | | | ref. | | |
| Middle school | 1.27 | 1.14 | 1.42 | 1.14 | 1.05 | 1.24 | 1.12 | 1.03 | 1.22 |
| High school | 1.46 | 1.31 | 1.63 | 1.21 | 1.12 | 1.32 | 1.20 | 1.10 | 1.30 |
| Degree | 1.65 | 1.47 | 1.84 | 1.27 | 1.16 | 1.39 | 1.12 | 1.01 | 1.23 |
| **Economic difficulties** | | | | | | | | | |
| Many | ref. | | | ref. | | | ref. | | |
| Some | 1.05 | 0.99 | 1.12 | 1.08 | 1.00 | 1.16 | 1.27 | 1.16 | 1.38 |
| None | 1.20 | 1.12 | 1.28 | 1.27 | 1.18 | 1.37 | 1.74 | 1.59 | 1.89 |
| **Citizenship** | | | | | | | | | |
| Italian | | | | | | | | | |
| Foreigner | 0.95 | 0.89 | 1.01 | 0.81 | 0.73 | 0.90 | 0.99 | 0.87 | 1.12 |
| **Period** | | | | | | | | | |
| 2017–2019 | ref. | | | ref. | | | ref. | | |
| 2020 | 0.76 | 0.73 | 0.80 | 0.83 | 0.78 | 0.89 | 0.74 | 0.69 | 0.79 |



**Figure 5.** Trends of the proportion of the screening target population who declared to have had a test in due time, by geographical macro area. Data from the PASSI interviews. For breast cancer, we considered as being eligible the female population aged 50–69 years and those who reported as having had a mammogram in the last 2 years as up to date with screening; for cervical cancer, we considered as being eligible the female population aged 25–64 years and those having had a Pap test in the last 3 years or an HPV-DNA test in the last 5 years as up to date with screening; for colorectal cancer, we considered as being eligible males and females aged 50–69

*Figure 5 continued on next page*

*Figure 5 continued*

years and those who reported having had a fecal occult blood test (FOBT) in the last 2 years or a colonoscopy or sigmoidoscopy in the last 5 years as up to date with screening.

Multivariate Poisson models show that the probability of having a test in the last year was lower even when adjusting for all other variables (*Table 3*). Furthermore, stratifying the Poisson models by period, in 2020, the probability of having had a test in the last year showed larger differences according to the level of education than in the pre-pandemic period for the three screenings (*Table 4*); nevertheless, the differences could be due to random fluctuations. Furthermore, in 2020, for breast cancer screening only foreigners had a lower probability of having had a test than Italians, inverting what was observed in the pre-pandemic period (*Table 3*). The differences by age and economic difficulties remained substantially unchanged in the pandemic compared with the pre-pandemic period.

## Discussion

The interruption of screening programs during lockdown over March–May 2020, as well as the reduction in their activity in the following months, caused, on average, a delay of at least 6 months for

**Table 4.** Multivariable Poisson regression models comparing the prevalence of having had a test in the last year by age, sex, familial status, socioeconomic characteristics, and citizenship in the pandemic and pre-pandemic period for cervical, breast, and colorectal cancer screening in Italy.

| | Cervix | | | | | | Breast | | | | | | Colorectal | | | | | |
|---|---|---|---|---|---|---|---|---|---|---|---|---|---|---|---|---|---|---|
| | 2017–2019 | | | 2020 | | | 2017–2019 | | | 2020 | | | 2017–2019 | | | 2020 | | |
| | PRR | 95% CI | | PRR | 95% CI | | PRR | 95% CI | | PRR | 95% CI | | PRR | 95% CI | | PRR | 95% CI | |
| **Age (years)** | | | | | | | | | | | | | | | | | | |
| 25–34 | 1.05 | 1.00 | 1.10 | 1.11 | 0.96 | 1.27 | | | | | | | | | | | | |
| 35–49 | 1.10 | 1.06 | 1.14 | 1.10 | 0.98 | 1.24 | | | | | | | | | | | | |
| 50–64 | ref. | | | ref. | | | | | | | | | | | | | | |
| 50–59 | | | | | | | 1.13 | 1.08 | 1.18 | 1.14 | 1.00 | 1.29 | ref. | | | ref. | | |
| 60–69 | | | | | | | ref. | | | ref. | | | 1.15 | 1.10 | 1.20 | 1.15 | 1.01 | 1.31 |
| **Sex** | | | | | | | | | | | | | | | | | | |
| Male | | | | | | | | | | | | | ref. | | | ref. | | |
| Female | | | | | | | | | | | | | 0.98 | 0.94 | 1.02 | 1.01 | 0.89 | 1.15 |
| **Familial status** | | | | | | | | | | | | | | | | | | |
| Married or with partner | 1.09 | 1.05 | 1.13 | 1.10 | 0.99 | 1.22 | 1.04 | 0.99 | 1.09 | 1.01 | 0.88 | 1.15 | | | | | | |
| Alone | ref. | | | ref. | | | ref. | | | ref. | | | | | | | | |
| **Educational level** | | | | | | | | | | | | | | | | | | |
| No title/elementary | ref. | | | ref. | | | ref. | | | ref. | | | ref. | | | ref. | | |
| Middle school | 1.24 | 1.11 | 1.39 | 1.56 | 0.98 | 2.49 | 1.10 | 1.02 | 1.20 | 1.52 | 1.14 | 2.02 | 1.10 | 1.01 | 1.20 | 1.29 | 0.94 | 1.77 |
| High school | 1.41 | 1.26 | 1.58 | 1.89 | 1.19 | 3.00 | 1.17 | 1.08 | 1.27 | 1.62 | 1.21 | 2.17 | 1.18 | 1.08 | 1.29 | 1.37 | 1.00 | 1.89 |
| Degree | 1.58 | 1.41 | 1.77 | 2.21 | 1.38 | 3.54 | 1.23 | 1.12 | 1.36 | 1.61 | 1.18 | 2.21 | 1.06 | 0.96 | 1.17 | 1.47 | 1.04 | 2.09 |
| **Economic difficulties** | | | | | | | | | | | | | | | | | | |
| Many | ref. | | | ref. | | | ref. | | | ref. | | | ref. | | | ref. | | |
| Some | 1.06 | 1.00 | 1.13 | 1.03 | 0.82 | 1.29 | 1.08 | 1.00 | 1.17 | 1.04 | 0.81 | 1.34 | 1.27 | 1.16 | 1.39 | 1.27 | 0.95 | 1.70 |
| None | 1.20 | 1.13 | 1.28 | 1.18 | 0.94 | 1.47 | 1.27 | 1.17 | 1.37 | 1.29 | 1.00 | 1.67 | 1.72 | 1.58 | 1.88 | 1.83 | 1.38 | 2.42 |
| **Citizenship** | | | | | | | | | | | | | | | | | | |
| Italian | ref. | | | ref. | | | ref. | | | ref. | | | ref. | | | ref. | | |
| Foreigner | 0.93 | 0.87 | 0.99 | 1.06 | 0.88 | 1.27 | 0.83 | 0.74 | 0.92 | 0.72 | 0.52 | 1.01 | 0.93 | 0.82 | 1.05 | 1.31 | 0.94 | 1.82 |

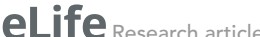

**Figure 6.** Trends of the proportion of the screening target population who declared to have had a test in due time, by education. Data from the PASSI interviews. For breast cancer, we considered as being eligible the female population aged 50–69 years and those who reported as having had a mammogram in the last 2 years as up to date with screening; for cervical cancer, we considered as being eligible the female population aged 25–64 years and those having had a Pap test in the last 3 years or an HPV-DNA test in the last 5 years as up to date with screening; for colorectal cancer, we considered as being eligible males and females aged 50–69 years and those who reported as having had a fecal occult blood test (FOBT) in the last 2 years or a colonoscopy or sigmoidoscopy in the last 5 years as up to date with screening. Educational attainment was groped in two categories: low (no title, elementary school, or middle school); high (high school or higher education).

cervical cancer, 5 months for breast cervical, and 6 months for colorectal cancer screening. There are large differences in the cumulated delay between macro areas and, within macro areas, between regions (*Table 2*) and local health authorities (*Mangone et al., 2022*; *Gathani et al., 2021*). The largest delays are observed in those areas where screening programs had historical problems in extending invitations to the whole target population and participation was already low before the pandemic – particularly in Southern Italy but also in some areas of Northern Italy – where cervical cancer screening was recently implemented and coverage relied largely on opportunistic screening (*Petrelli et al., 2018*; *Giorgi Rossi et al., 2019*; *Giorgi Rossi et al., 2018*). Northern Italy was also the most affected area by the pandemic.

It is worth noting that the decrease in screening tests performed by screening programs was larger than the decrease in invitations. Even if the surveys conducted by the National Screening Monitoring Center were not designed to measure participation, this difference in the decrease indirectly shows that participation decreased during the study period.

Stopping screening programs and their slow restart caused an appreciable decrease in test coverage in the target population of breast and colorectal cancer. This decrease is smaller, as expected, for cervical cancer screening because the longer screening intervals reduce the impact of the period of absence or reduced activity; nevertheless, a change in the direction of the trend is also appreciable for cervical cancer screening. While for colorectal screening the contribution of opportunistic screening was negligible before and during the pandemic, for breast and cervical cancer opportunistic screening did not increase the proportion of population test coverage and only a small peak of women reported having paid for a test was appreciable in the strict lockdown period of March–May 2020.

The decrease in test coverage provided by organized screening programs caused an increase in inequalities. In fact, people with a lower level of education and immigrants paid the largest lack of access to secondary prevention during the pandemic.

Other studies reported an early disruption of screening activities following the lockdown, with invitations and first-level tests being stopped, and a reduction in participation when invitations restarted (*Eijkelboom et al., 2021a*; *Peacock et al., 2021*; *Ho et al., 2022*; *Eijkelboom et al., 2021b*; *Bosch et al., 2022*).

The reported data show large differences across countries in the screening programs' ability to resume their activity and in catching up with the cumulated backlog. Italy has a federal health system in which the implementation of screening programs is delegated to the regional government and practically managed by the local health authorities. This organizational model, together with historical differences in the robustness of screening programs and the population's trust in the public health system, resulted in an extreme variability in the delay cumulated in more than 1 year of COVID-19 emergency (*Giorgi Rossi et al., 2019*). In fact, some areas showed the ability to recover all the backlog, while the vast majority were still cumulating further delay in the first months of 2021. These differences increased the already existing geographical inequalities across the country.

Therefore, individual inequalities are also going to increase. In fact, the difference by educational level seems to be stronger in 2020 than in previous years. Furthermore, in breast cancer screening differences disadvantaging immigrants – that were small in previous years – became larger in 2020. Even if this difference could be due to chance, it may also reflect that immigrants rely mostly on organized screening and scarcely on opportunistic screening for mammography, which is a relatively expensive test. Studies from the US also showed increased inequalities consequent to the screening program interruption, with a larger impact in the decrease of screening uptake in rural areas and for beneficiaries of public insurance or those who are not insured at all (*Monsivais et al., 2022*, *Amram et al., 2022*).

## Possible impact

Many studies from Italy and other countries reported a delay in diagnoses for many cancer sites (*Vanni et al., 2020*, *Gathani et al., 2021*). In some studies, a shift to more advanced stages and different initial therapeutic approaches have been observed for breast cancer and colorectal cancers (*Toss et al., 2021*; *Vanni et al., 2021*; *Vives et al., 2022*; *Blay et al., 2021*; *Longcroft-Wheaton*



**Figure 7.** Trends of the proportion of the screening target population who declared to have had a test in due time, by economic difficulties, Data from the PASSI interviews. For breast cancer, we considered as being eligible the female population aged 50–69 years and those who reported as having had a mammogram in the last 2 years as up to date with screening; for cervical cancer, we considered as being eligible the female population aged 25–64 years and those having had a Pap test in the last 3 years or an HPV-DNA test in the last 5 years as up to date with screening; for colorectal cancer, we considered as being eligible males and females aged 50–69 years and those who reported as having had a fecal occult blood test (FOBT) in the last 2 years or a colonoscopy or sigmoidoscopy in the last 5 years as up to date with screening. Economic difficulties are classified into three categories: many economic difficulties; some economic difficulties; no economic difficulties.

**Figure 8.** Proportion of the target population who declared having had the screening test in the last year, by year and setting where the test was last performed. Data from the PASSI interviews.

*et al., 2021*). Investigating the impact on the cancer stage is out of the scope of this study. Nevertheless, computing the expected delay cumulated up to now can give an estimate of the impact on mortality and, for cervical and colorectal cancer, on incidence. In fact, several mathematical models have been adapted precisely for this scope. For breast and colorectal cancer, in England, a model assuming a 12-month suspension of screening and early diagnosis pathways and reallocating all diagnoses to symptomatic diagnosis estimated an excess of about 300 breast cancer deaths (8–10% increase) and 1500 colorectal cancer deaths (15–17%) in the next 5 years (*Maringe*

*et al., 2020*). The expected health impact of the disruption may be larger for clinical than for screening services. The results of simulation models focused on the analysis of the impact of screening programs disruption are suggesting that we can expect a relative increase in breast and colorectal cancer-specific mortality ranging between 1% and 3% over the next 10–30 years, depending on the duration of the disruption and on the catch-up strategies adopted. More than half of the excess deaths are expected to occur during the first 5–10 years following disruption and the health impact might be larger for older people and disadvantaged population subgroups. For cervical cancer, it has been estimated that a delay of 6 months national screening program would lead to about 600 more cancers in England that would occur in the next screening round in the absence of catch-up strategies (*Castanon et al., 2021*, *Castanon et al., 2022*). We can expect a similar impact of screening disruption in Italy, where we observed a wide variability in the length of disruption, with a 6-month average delay in the invitations (*Kregting et al., 2021*; *de Jonge et al., 2021*; *Duffy et al., 2022*).

## Conclusions

The lockdown and the ongoing COVID-19 emergency caused an important delay in screening activities. Catch-up of backlog was different across regions, and differences cannot be explained by the severity of the pandemic in different areas. The resilience of the screening programs seems to reflect the historical robustness of the organization with areas that were able to reach higher invitation and test coverage reacting more promptly to the COVID-19 crisis. The delay of screening programs increased the preexisting individual and geographical inequalities in access. The opportunistic screening did not mitigate the pandemic impact.

## Acknowledgements

The authors are grateful to all the regional and local coordinators and interviewers of PASSI surveillance and to the regional screening coordinators, who contributed to the data collection. A special thanks goes to the PASSI group for their competence and commitment. This study was partially supported by the Italian Ministry of Health – Ricerca Corrente Annual Program 2023.

## Additional information

### Funding

| Funder | Grant reference number | Author |
| --- | --- | --- |
| Ministero della Salute | Ricerca corrente 2023 | Paolo Giorgi Rossi |

The funders had no role in study design, data collection and interpretation, or the decision to submit the work for publication.

### Author contributions

Paolo Giorgi Rossi, Conceptualization, Formal analysis, Funding acquisition, Methodology, Writing - original draft; Giuliano Carrozzi, Data curation, Formal analysis, Methodology, Writing – review and editing; Patrizia Falini, Paola Armaroli, Data curation, Investigation, Writing – review and editing; Letizia Sampaolo, Maria Masocco, Data curation, Supervision, Writing – review and editing; Giuseppe Gorini, Data curation, Supervision, Investigation, Writing – review and editing; Manuel Zorzi, Data curation, Writing – review and editing; Carlo Senore, Priscilla Sassoli de Bianchi, Writing – review and editing; Marco Zappa, Conceptualization, Methodology, Writing – review and editing; Francesca Battisti, Supervision, Writing – review and editing; Paola Mantellini, Conceptualization, Resources, Supervision, Investigation, Methodology, Writing – review and editing

### Author ORCIDs

Paolo Giorgi Rossi ⓘ http://orcid.org/0000-0001-9703-2460
Francesca Battisti ⓘ http://orcid.org/0000-0001-5778-5820
Paola Mantellini ⓘ http://orcid.org/0000-0003-4114-4011

### Ethics

Human subjects: Screening activity is monitored by ONS as statutory duties on regular basis, using a standard common set of quality indicators. During the Covid 19 pandemic ONS conducted this analysis as a part of the routine monitoring activity of the programmes performance, pooling anonymous individual data from each programme, based on a common standardised form. Approval from local ethics review boards is not required for monitoring programme activity. Regarding PASSI surveillance system, personal data are processed in compliance with the GDPR 2016. PASSI was approved by the Ethics Committee of the National Institute of Public Health on January 23, 2007. Interviews are transferred anonymously to a national archive via a secure internet connection. Personal Identifiers on paper or computers are subsequently locally destroyed.

### Decision letter and Author response

Decision letter https://doi.org/10.7554/eLife.81804.sa1
Author response https://doi.org/10.7554/eLife.81804.sa2

---

## Additional files

### Supplementary files

• Supplementary file 1. ONS dataset. Cervical cancer screening invitation and examination coverage 2017–2020 in Italy.

• Supplementary file 2. ONS dataset. Breast cancer screening invitation and examination coverage 2017–2020 in Italy.

• Supplementary file 3. ONS dataset. Colorectal cancer screening invitation and examination coverage 2017–2020 in Italy.

• Supplementary file 4. PASSI dataset. Cervical cancer screening: trend of the proportion of the screening target population who declared to have had a test in due time, overall and by setting of the last test, geographical macro area, education, and economic difficulties.

• Supplementary file 5. PASSI dataset. Breast cancer screening: trend of the proportion of the screening target population who declared to have had a test in due time, overall and by setting of the last test, geographical macro area, education, and economic difficulties.

• Supplementary file 6. PASSI dataset. Colorectal cancer screening: trend of the proportion of the screening target population who declared to have had a test in due time, overall and by setting of the last test, geographical macro area, education, and economic difficulties.

• Supplementary file 7. Number of interviews included in the analysis of PASSI's survey.

• MDAR checklist

### Data availability

The study reports the results of mandatory monitoring activities, that are statutory duties of the National Screening Monitoring System (ONS). Although the anonymized dataset is not yet available, ONS is working to make it available as open data on its website. In the PASSI surveillance system, personal data are processed in compliance with the GDPR 2016. Although the anonymized dataset is not yet available, the National Institute of Public Health is working to make it available on request (http://www.epicentro.iss.it/passi/PresPolicy.asp) and the excel sheets with the numbers used to plot the graphs and charts of the manuscript are available and enclosed as supplementary files.

---

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
