## [Editor Report]

The authors provide important evidence for the impact of the COVID-19 pandemic on screening for breast, cervix, and colorectal cancer in Italy. They compared Invitation and examination coverage, conducted telephone interviews, and investigated screening test coverage, before and during the pandemic, according to educational attainment, perceived economic difficulties and citizenship. Their findings convincingly show that the lockdown and pandemic restrictions caused delays in screening and increased the pre-existing individual and geographical inequalities in access.

---

## [Decision Letter]

**Decision letter after peer review:**

Thank you for submitting your article "The impact of the COVID-19 pandemic on Italian population-based cancer screening activities and test coverage: results from national cross-sectional repeated surveys" for consideration by *eLife*. Your article has been reviewed by 3 peer reviewers, and the evaluation has been overseen by me in a dual role of Reviewing Editor and Senior Editor. The following individuals involved in review of your submission have agreed to reveal their identity: Jonine Figueroa (Reviewer #1); James F O'Mahony (Reviewer #3).

Essential revisions:

As is customary in *eLife*, the reviewers have discussed their critiques with one another and with the Reviewing and Senior Editor. The decision was reached by consensus. What follows below is an edited compilation of the essential and ancillary points provided by reviewers in their critiques and in their interaction post-review. Please submit a revised version that addresses these concerns directly. Although we expect that you will address these comments in your response letter, we also need to see the corresponding revision clearly marked in the text of the manuscript. Some of the reviewers' comments may seem to be simple queries or challenges that do not prompt revisions to the text. Please keep in mind, however, that readers may have the same perspective as the reviewers. Therefore, it is essential that you amend or expand the text to clarify the narrative accordingly.

*Reviewer #1 (Recommendations for the authors):*

– The presentation of Box1 describing the screening guidelines is useful--should this be a table?

– The authors describe some of the differences between regions, it might be helpful to have a map figure describing the different regional boards and their differences in Covid rates/lockdowns during the study period or perhaps a timeline of the different mitigation measures. Not clear what the 21 regions are and confusing when subsequently state that for Calabria 2/5 missing--I am assuming there's a lower subset beyond region?

– The figures quality are poor and were hard for me to read although I could determine the information being presented. Also for Figure 1 it was unclear to me how is it that coverage is over 100% in the central region, presumably this is because of the catch up program but perhaps the timing of this needs to be separated as opposed to showing it in one bar? Why was it that Central region was able to catch up compared to other regions? Is there lessons to be learned here?

*Reviewer #3 (Recommendations for the authors):*

I enjoyed reading this well-written analysis. Generally the standard of academic writing in this manuscript is very high. I see some scope for adjustment to more natural English is some cases.

L27

Stating that screening is mandatory implies the population is required to screen whereas the requirement is presumably on the local health system to offer screening to all those who wish to receive it. Some change in language might be necessary.

L29

I believe COVID-19 should be capitalised.

L34-35

Sentence requires revision for clarity.

L37

among -> between

L39/40

"programmes" might be more natural than "campaigns". A programme is an ongoing service while a campaign is associated with a short term push to improve participation.

L53

I would state that the COVID pandemic emerged in early 2020 as readers in the future might benefit from this context.

L58-64

The thematic flow of this paragraph isn't quite clear. All of the individual elements are certainly relevant, but how they fit together is not.

L70

participation to -> participation in

L77-79

I would not mix fractions and percentages in one sentence.

L97

Clarify you mean the first COVID case (ie not cancer). I know this should be clear from the context but I think it is important to be specific.

L97-123

This is a very long paragraph. I suggest breaking into three.

Also, this material is more or less a description of events not a description of research methods. There is no problem with the material itself, but I think it should be placed into the introduction described as background context or similar.

L105

If follow-up was never stopped nationally (ie all regions), please state explicitly.

L110

Do you have a source to cite for the reduction by 30-50%? If there is no source can you indicate that this information might be less formal.

L125

yy-> yrs for ages, yy -> yl for yearly.

L144

PASSI -> The PASSI / PASSI's survey

Figure 1 is a good illustration of the data and easy to interpret. Would it be better to place Figures 1 and 3 and 2 and 4 side by side and using the same vertical scale allowing the reader to easily compare the trends in invitations and coverage?

All Figures are of poor resolution. I understand this may be due to the journal's submission portal rather than the file supplied by the authors but it would look better if produced at a greater resolution.

I suggest "Nationwide" or "National Average" instead of "Italy" as a geographical category. In Figures 2/4 I would place the macro region below the bar plot rather than over it.

Colonrectum -> Colorectal

Table 1, use consistent one decimal place.

In the Figure titles for Figures 1-5 I think it would be good to make clear early in the title if the data source is ONS or PASSI to help readers interpret if this is programme data or self-report of individuals from surveys. For example, the title in Figure 5 only clarifies that this is PASSI data in the last sentence of the title.

The colours in Figure 5 are not very different for Breast and Cervix. The colours chosen for colorectal are easier to tell apart. Consider how these figures will look if printed in black and white.

I would use the same 0-100% vertical scale for all three plots in Figure 5 to make the absolute difference between them clear.

The figure title for Figure 7 needs to correspond more closely with the high/low label used in the legend to make it clear what this refers to.

Similarly, for Figure 8 the groups described in the title do not clearly correspond to what is described in the legend.

Table 2

Economic difficulties: no-> none

I think some greater interpretation of Table 2 would be beneficial for the reader within the Results section.

L382

Can we state anything meaningful about the screening for foreigners in cervix and colorectal if the CIs cross one?

---

## [Author Response]

Reviewer #1 (Recommendations for the authors):– The presentation of Box1 describing the screening guidelines is useful--should this be a table?

We changed the title in table 1.

– The authors describe some of the differences between regions, it might be helpful to have a map figure describing the different regional boards and their differences in Covid rates/lockdowns during the study period or perhaps a timeline of the different mitigation measures. Not clear what the 21 regions are and confusing when subsequently state that for Calabria 2/5 missing--I am assuming there's a lower subset beyond region?

We added a figure reporting the maps of incidence and mortality, during the first and second pandemic waves, by province and showing how the country is divided into three macro areas. We added a sentence to explain how screening activities data are collected, this should make clear that, in some regions, the screening program is only one and collects regional data, while in other regions the programs are organized at the local health authority level and data are collected separately for each program.

– The figures quality are poor and were hard for me to read although I could determine the information being presented. Also for Figure 1 it was unclear to me how is it that coverage is over 100% in the central region, presumably this is because of the catch up program but perhaps the timing of this needs to be separated as opposed to showing it in one bar? Why was it that Central region was able to catch up compared to other regions? Is there lessons to be learned here?

The definition of invitation coverage is “the percentage of citizens who were sent an invitation to a screening during the analyzed period, compared to the population to be invited in the period in order to reach all the target population in the screening interval, excluding non-eligible subjects”. Therefore it is possible that a program invites more than the expected target population in a certain period. This occurred also after the lockdown in order to catch up with the backlog.

It is difficult to understand why some programs were able to catch up and others are still facing delays. It was clearly not linked to the severity of the pandemic, but it is linked to the program’s ability to cover the population before Covid-19. We tried to expand on this point in the conclusions:

“The lockdown and the ongoing COVID-19 emergency caused an important delay in screening activities. Catch-up of backlog was different across regions, differences cannot be explained by the severity of the pandemic in different areas. The resilience of the screening programs seems to reflect the historical robustness of the organization with areas that were able to reach higher invitation and test coverage reacting more promptly to the COVID-19 crisis. The delay of screening programs increased the pre-existing individual and geographical inequalities in access. The opportunistic screening did not mitigate the pandemic impact.”

Reviewer #3 (Recommendations for the authors):I enjoyed reading this well-written analysis. Generally the standard of academic writing in this manuscript is very high. I see some scope for adjustment to more natural English is some cases.

We thank the reviewer for this encouraging comment.

L27Stating that screening is mandatory implies the population is required to screen whereas the requirement is presumably on the local health system to offer screening to all those who wish to receive it. Some change in language might be necessary.

We reworded the sentence. The reviewer is correct: the regional health systems must implement screening programs with the active invitation of the population.

L29I believe COVID-19 should be capitalised.

Thanks, done.

L34-35Sentence requires revision for clarity.

We rewrote the sentence as follows: “PASSI surveillance system, through telephone interviews, collects information about screening test uptake by test provider (public screening and private opportunistic). Test coverage and test uptake in the last year were computed, by educational attainment, perceived economic difficulties and citizenship.”

L37among -> between

Thanks, done.

L39/40"programmes" might be more natural than "campaigns". A programme is an ongoing service while a campaign is associated with a short term push to improve participation.

Thanks, these are definitely programs and not campaigns!

L53I would state that the COVID pandemic emerged in early 2020 as readers in the future might benefit from this context.

Thanks, done.

L58-64The thematic flow of this paragraph isn't quite clear. All of the individual elements are certainly relevant, but how they fit together is not.

We changed the sentence to explain the consequentiality of three concepts:

“Cancer screening tests are non-urgent services and thus they were among the first suspended activities during the first pandemic wave in most European countries. On the other hand, organised screening programmes actively invite the target population. The active invitation allows to accurately plan the workload, which represented an opportunity for organised screaming programmes to resume post-lockdown activities in a rational way according to accurate prioritisation, aiming to minimise the impact of the pandemic on cancer diagnosis delays. Thus, the presence of a structured and well-organised programme has been recognised as a possible element favouring the resilience of health services to the pandemic disruption.

L70participation to -> participation in

Thanks, done.

L77-79I would not mix fractions and percentages in one sentence.

We rewrote the paragraph splitting it into two sentences. We left the percentage and fractions because the second ones referred only to the covered population.

L97Clarify you mean the first COVID case (ie not cancer). I know this should be clear from the context but I think it is important to be specific.

Thanks, done.

L97-123This is a very long paragraph. I suggest breaking into three.Also, this material is more or less a description of events not a description of research methods. There is no problem with the material itself, but I think it should be placed into the introduction described as background context or similar.

We made a separate subheading on the pandemic. This is part of the background but also a setting description. We prefer to maintain it in the methods in a separate subheading and maintain the introduction shorter in order to enunciate the objectives of the study before the detailed description of the context. We moved the description of the changes in screening program organization in the results since this information has been collected with the ad hoc survey by the ONS. The remaining paragraph is much smaller and, we hope, easier to read.

L105If follow-up was never stopped nationally (ie all regions), please state explicitly.

It is correct that all assessment tests were never stopped or delayed (at least in theory, practically some delays occurred because of a lack of anesthetists where needed or patients not presenting or changing the appointments). About oncologic follow-up and post-treatment follow-up of pre-cancerous lesions, this was the directive, but this function is not always managed by the screening program, thus in some cases, we do not know how these visits have been managed. In this paragraph, we report the national recommendations.

L110Do you have a source to cite for the reduction by 30-50%? If there is no source can you indicate that this information might be less formal.

The source of information is the survey itself. A short questionnaire on the changes in the program was included. We reported this in the methods and moved all the information collected through the survey.

L125yy-> yrs for ages, yy -> yl for yearly.

Thanks, done.

L144PASSI -> The PASSI / PASSI's survey

Thanks, done.

Figure 1 is a good illustration of the data and easy to interpret. Would it be better to place Figures 1 and 3 and 2 and 4 side by side and using the same vertical scale allowing the reader to easily compare the trends in invitations and coverage?

We thank the reviewer for this suggestion. We changed the figures accordingly.

All Figures are of poor resolution. I understand this may be due to the journal's submission portal rather than the file supplied by the authors but it would look better if produced at a greater resolution.

We improved the quality of the figures. Nevertheless, figures embedded in a word file cannot have a good resolution. For the final version, if accepted, we will give the original files.

I suggest "Nationwide" or "National Average" instead of "Italy" as a geographical category. In Figures 2/4 I would place the macro region below the bar plot rather than over it.

Thanks, done.

Colonrectum -> Colorectal

Thanks, done.

Table 1, use consistent one decimal place.

Thanks, done.

In the Figure titles for Figures 1-5 I think it would be good to make clear early in the title if the data source is ONS or PASSI to help readers interpret if this is programme data or self-report of individuals from surveys. For example, the title in Figure 5 only clarifies that this is PASSI data in the last sentence of the title.

We changed the figure legends identifying a figure title, where we report the source of information and the text of the legend.

The colours in Figure 5 are not very different for Breast and Cervix. The colours chosen for colorectal are easier to tell apart. Consider how these figures will look if printed in black and white.

Thanks, done.

I would use the same 0-100% vertical scale for all three plots in Figure 5 to make the absolute difference between them clear.

Thanks, done.

The figure title for Figure 7 needs to correspond more closely with the high/low label used in the legend to make it clear what this refers to.

Thanks, done.

Similarly, for Figure 8 the groups described in the title do not clearly correspond to what is described in the legend.

We thank the reviewer, there was a mistake, and we changed it accordingly.

Table 2Economic difficulties: no-> none

Thanks, done.

I think some greater interpretation of Table 2 would be beneficial for the reader within the Results section.

We better explained what is reported in table 3 (former table 2).

L382Can we state anything meaningful about the screening for foreigners in cervix and colorectal if the CIs cross one?

We agree with the reviewer, even if we did not adopt a significance threshold for accepting or refusing the null hypothesis, changes in cervical and colorectal cancers are not in the direction discussed for breast cancer. Sorry for this mistake, we changed the sentence accordingly.